# Effects of Denture Treatment on Salivary Metabolites: A Pilot Study

**DOI:** 10.3390/ijms241813959

**Published:** 2023-09-11

**Authors:** Narumi Ichigaya, Norishige Kawanishi, Takuya Adachi, Masahiro Sugimoto, Katsuhiko Kimoto, Noriyuki Hoshi

**Affiliations:** 1Department of Fixed Prosthodontics, Kanagawa Dental University, Yokosuka 238-8580, Japan; narumininatta@gmail.com (N.I.); kawanishi@kdu.ac.jp (N.K.); adachi@kdu.ac.jp (T.A.); k.kimoto@kdu.ac.jp (K.K.); 2Institute of Medical Sciences, Tokyo Medical University, Shinjuku 160-8402, Japan; mshrsgmt@gmail.com; 3Institute for Advanced Biosciences, Keio University, Tsuruoka 997-0052, Japan; 4Department of Education Planning, Kanagawa Dental University, Yokosuka 238-8580, Japan

**Keywords:** oral discomfort, denture treatment, metabolomics, metabolites, resting saliva, stimulated saliva

## Abstract

Symptoms of oral discomfort such as dry mouth are common in older people wearing dentures. Such symptoms are mainly treated symptomatically. Many of these symptoms are related to saliva, and associations with salivary volume have been reported. Although denture treatment improves symptoms by increasing the amount of saliva, the effects on salivary components remain unclear. This study aimed to investigate the effects of denture treatment on salivary metabolite changes based on salivary metabolome analyses. We enrolled 21 patients requiring denture treatment. At the first visit, and after completion of denture treatment, saliva outflow was measured under resting and stimulated conditions, samples for salivary metabolite analysis were collected, and masticatory efficiency was tested. In all participants, masticatory efficiency increased after denture treatment. Moreover, the amounts of resting and stimulated saliva were increased. Using salivary metabolome analysis, 61 salivary metabolites were detected. Substantial concentration changes were observed for 4 and 21 metabolites in resting and stimulated saliva, respectively. The four metabolites common to both saliva tests had significantly lower concentrations after treatment. These results suggest that the improvement in masticatory function by dentures is related not only to salivary secretion volume, but also to salivary metabolite composition.

## 1. Introduction

An increasing number of older people who use dentures experience symptoms of oral discomfort, such as dry mouth [1], and more than half of patients with dry mouth are older individuals [2]. The main causes of dry mouth are Sjögren’s syndrome [3], salivary gland dysfunction following radiotherapy or surgical resection [4], unwanted effects of medications [5], metabolic diseases such as diabetes mellitus [6,7,8], and ageing [9]. However, no clear cause was identified [10]. The general treatment for dry mouth is symptomatic, a “wait-and-see” approach, moisturization, and medication to improve symptoms [11]. However, dry mouth is often difficult to manage in older people because it is caused or exacerbated by medications for their underlying diseases [12].

Dentures are a risk factor for oral disease in the elderly. Denture-related diseases, such as stomatitis, angio-auricular stomatitis, ulcers, and hyperplasia, are known to be oral-cavity-related lesions [13]. One of the most significant factors causing oral symptoms is the use of ill-fitting dentures. Tooth loss reduces quality of life, and rehabilitation with dentures can prevent deterioration of quality of life [14]. Denture treatment is not only effective for aesthetic and functional recovery, such as mastication, but also affects the patient’s general condition as the use of dentures improves the nutritional intake efficiency and thereby the patient’s nutritional status [15]. Patients’ satisfaction with prosthetic denture treatment are reportedly related to antioxidants and oxidizing enzymes in saliva [16]. However, ill-fitting dentures not only result in mucosal lesions in the oral cavity, they also lead to oral discomfort symptoms, nutritional intake, and social communication problems [17]. Candida albicans is the focus of attention in relation to dentures and oral symptoms. There are differences in the prevalence of oral candidiasis between denture wearers and non-denture wearers, with a higher prevalence reported in denture wearers [18].

Ill-fitting dentures may induce oral symptoms and oral candidiasis. A decrease in saliva flow rate is associated with these symptoms. With a focus on the salivary flow rate, investigators reported changes in oral symptoms and salivary flow rate in denture wearers after new dentures were made or repaired. As a result of denture improvements, an improvement in salivary flow rate and alleviation of oral symptoms were observed [1]. Furthermore, it was reported that the effect of changes in salivary flow rate during stimulated saliva was strong. In a similar study, improvements in saliva flow rate and improvements in dentures in denture wearers suffering from oral candidiasis were reported to affect saliva flow rate at the time of irritation as a factor [19]. Based on these changes, dentures and saliva, as factors related to oral symptoms are potentially closely related.

Saliva contains various components, such as substances transferred from the blood to the salivary glands and those derived from oral bacteria [20]. Recently, saliva tests have been used for COVID-19 antigen tests and PCR kits. These are widely used as relatively simple and less invasive methods [21]. In clinical dentistry, saliva tests are also used to assess the risk of dental caries and periodontal disease, as well as tests for bad breath. Furthermore, the amount of secreted saliva is clinically used as an indicator of various diseases. Decreased salivary secretion increases the risk of dental caries and periodontal disease, causes pain, affects oral functions such as eating and swallowing, and increases the risk of various infections and pneumonia [22]. Salivary secretion is also an index of systemic quality of life and an important parameter for evaluating oral health conditions [23].

Saliva tests are used to evaluate not only the amount of saliva secreted, but also salivary components based on various methods [24]. Recent advances in science and technology have drawn attention to omics analysis, which comprehensively investigates various molecules that constitute the living body. Metabolomics analyzes metabolites in vivo to clarify the relationships between metabolic changes, biological functions, and various pathologies [25]. Saliva can be used as a specimen in such tests, and salivary metabolome analyses detected changes in patients with periodontal disease or diabetes mellitus [26,27] Metabolomic biomarkers’ use for oral cancer screening is currently being explored, and earlier detection is expected [28]. The possibility of early detection by metabolomic biomarkers has also been reported for oral candidiasis, which is related as an oral symptom [29]. As for oral functions, attention has been paid to mastication, and differences in salivary metabolites between stimulated saliva and resting saliva have been reported [30]. However, it remains unclear whether denture treatment affects salivary metabolites that reflect the patient’s general condition.

We investigated salivary metabolites, in addition to the amount of salivary secretion under resting and stimulated conditions, which have previously been found to change with denture treatment. This study aimed to investigate the effects of dentures on occlusal improvements in metabolites in resting and stimulated saliva using capillary electrophoresis time-of-flight mass spectrometry (CE-TOFMS). The null hypothesis is that the effect of denture treatment on salivary metabolites leads to (1) no effect on resting saliva, and (2) no effect on stimulated saliva.

## 2. Results

### 2.1. Salivary Flow Rates

The amounts of saliva secreted under resting and stimulated conditions were compared (Figure 1). The amounts of secreted resting saliva before (3.17 ± 2.87 mL) and after (4.88 ± 2.99 mL) treatment were significantly different (*p* < 0.0001; Figure 1a). Many participants who were below the reference value of 1.5 mL/15 min before treatment exceeded the reference value after treatment, indicating an improvement.

Likewise, the amounts of stimulated saliva before (12.84 ± 7.26 mL) and after (14.07 ± 5.73 mL) treatment were also significantly different (*p* < 0.05; Figure 1b). Many participants who were below the reference value of 10 mL/10 min before treatment showed an improvement above the reference value after the treatment. However, some participants whose values greatly exceeded the reference value before the treatment showed decreases after treatment to values within the reference range but not below the reference threshold.

### 2.2. Masticatory Ability

A masticatory efficiency test after saliva sampling was performed to evaluate the effectiveness of denture treatment. According to these results, many participants who were below the reference value of ≤100 mg/dL before treatment either exceeded the reference value or improved to the reference value after treatment, indicating a significant difference (Figure 2). In addition, those who exceeded the reference value before treatment showed a significant increase after the treatment (*p* < 0.0001).

### 2.3. Metabolome Analysis

#### 2.3.1. Comparison of Metabolites between Resting and Stimulated Saliva

The quantified metabolites were clustered and analyzed, and metabolites with large concentration changes were visualized as a heatmap (Figure 3). A total of 61 salivary metabolites were detected in the collected saliva samples, and differences in metabolite concentrations between resting and stimulated saliva were confirmed for all 61 metabolites. Clusters showing changes before and after treatment were found in clusters common to both the resting and stimulated saliva.

#### 2.3.2. Principal Component Analysis

PCA score and loading plots are shown in Figure 4a and b, respectively. Each point on the score plots represents one saliva sample, and a short point-to-point distance indicates a strong similarity in the patterns of salivary metabolite concentrations. Each point on the loading plots represents one metabolite that contributes to the principal component (PC). On the score plot, many points are distributed in the upper region (PC2 > 0) for resting saliva, whereas many points are accumulated in the lower region (PC2 < 0) for stimulated saliva.

PC1 is orthogonal to PC2, suggesting that changes occurred in some metabolites rather than in the concentration of all salivary metabolites (Figure 4a). Many metabolites were observed in the loading plots in regions with PC1 < 0 (Figure 4b). The contribution of PC1 (35.1%) was higher than that of PC2 (16.9%). PC1 reflects the overall concentration in the saliva samples, suggesting that the overall concentration increased as the value of PC1 increased. Comparing each saliva sample before and after treatment, the concentration pattern distribution area of the score plots changed for both saliva samples.

#### 2.3.3. Ratio of Metabolite Concentrations before and after Treatment

The volcano plot in Figure 5 shows the ratio of changes in metabolites before and after treatment in resting and stimulated saliva. In resting saliva samples, four metabolites (N1-acetyl spermidine, betaine, malate, and 2-hydroxy-4-methyl pentanoate) had significantly decreased concentrations after treatment (Figure 5a). In the volcano plot of stimulated saliva (Figure 5b), 17 substances (choline, N-acetyl aspartate, valine, 2-isopropyl malate, creatinine, carnitine, guanosine, alanine, butyrate, 2-hydroxy pentanoate, isoleucine, leucine, serine, lysine, adenosine, citrulline, and O-acetylcarnitine) showed a significant decrease in concentration after treatment, in addition to the four substances identified in resting saliva. None of the metabolites showed a significant concentration increase after treatment in either resting or stimulated saliva.

## 3. Discussion

In this study, we investigated the effects of denture treatment for occlusal improvement on salivary metabolites. The masticatory efficiency test, an objective evaluation of masticatory ability, showed increased values in all participants, demonstrating the effectiveness of the treatment. After the treatment, salivary volumes were increased under both resting and stimulated conditions. In total, 61 metabolites were detected in saliva samples. The concentrations of 4 and 21 metabolites in resting and stimulated saliva samples, respectively, showed marked changes before and after treatment. Significant differences were observed for the four substances common to both saliva tests.

Previous studies confirmed differences in salivary metabolites between resting and stimulated saliva [26]. In the present study, we also found differences in metabolites between resting and stimulated saliva, with more metabolites identified in stimulated than in resting saliva.

In this study, significant differences were observed in the amount of resting and stimulated saliva secretion before and after treatment. This result may have been caused by the improvement in masticatory function following denture treatment, which brought both saliva levels closer to the standard values. However, some participants showed a decreased amount of stimulated saliva. It is highly likely that in these patients, old dentures did not fit properly, resulting in excessive saliva flow, and appropriate denture treatment may have stabilized saliva secretion.

These findings suggest that denture treatment improved salivary secretion above the standard value and may affect changes in salivary metabolites. It has been suggested that masticatory function improvement following occlusal improvement due to prosthetic restoration may affect salivary metabolites. Our metabolome analysis demonstrated a significant decrease in four substances in resting saliva.

Among these, accumulation of N1-acetyl spermidine is associated with cell death [31,32], and denture treatment may have ameliorated the accumulation.

Betaine is an uncharged intramolecular salt that has inhibitory, cleansing and removing effects on biofilms, and also has anti-inflammatory effects [33]. The decrease in betaine concentration is thought to be caused by the reduction in biofilm due to the wearing of new dentures.

Malate, a metabolic intermediate of the citric acid cycle, increases upon stimulation with reactive oxygen species. The decrease in its concentration observed in our study may indicate an improvement in oral conditions [34,35].

2-Hydroxy-4-methyl pentanoate is an oxidoreductase that catalyzes the reduction in nicotinamide adenine dinucleotide, an electron carrier in bacteria. The observed decrease in 2-hydroxy-4-methyl pentanoate in stimulated saliva may have been due to the suppression of the amount and activity of bacteria following appropriate denture treatment [36].

Stimulated saliva showed significantly decreased concentrations of another 17 substances in addition to those also detected in resting saliva. In the future, it will be necessary to further investigate each component, focusing on the relationships between the four substances whose concentrations were significantly decreased and the effects of denture treatment.

Based on the results of this study, the new denture treatment has the following effects:(1)An improvement in oral condition by improvements in saliva volume.(2)It was thought that each metabolite improved as a biological defense reaction to the ill-fitting denture.

Salivary secretion also decreases with age [9]. In a previous study, a comparison of salivary metabolites showed the efficacy of stimulated saliva as well as the difference from resting saliva [30]. However, it did not provide sufficient knowledge as data were only for the target age group of this study. Further research in older people [37] who may be additionally affected by salivary gland atrophy is necessary.

Hypertension was a common systemic disease among the patients studied in this study. Therefore, we investigated the effect of taking antihypertensive drugs on salivary metabolites. A comparison of metabolites in subjects who had a history of taking medication for hypertension and subjects who had no history of taking medication did not reveal any effect on metabolites or salivary secretion. The reason for this is thought to be that the subjects who were taking antihypertensive drugs continued to take the drugs for a long period of time, and that the drugs were not changed during the study period. In the future, it will be necessary to fully examine target systemic diseases and examine their effects.

### Study Limitations

The participants in this study were limited to patients receiving new dentures to replace ill-fitting old dentures. Data from age-matched participants who did not need to wear dentures were not collected and therefore cannot be compared to a healthy population. Moreover, unlike previous studies, this study did not investigate the effects of participants’ age. We plan to conduct these studies in the future.

In this study, metabolites in resting saliva before and after denture treatment were identified as changing in concentration. As a result, four of the metabolites detected in resting saliva were affected by denture treatment. Thus, (1) the null hypothesis that denture treatment does not affect salivary metabolites in resting saliva was rejected. In addition, changes in the concentrations of metabolites in stimulated saliva were confirmed. The results showed that 21 of the metabolites detected in stimulated saliva were affected by denture treatment before and after denture treatment. Therefore, (2) the null hypothesis that denture treatment does not affect salivary metabolites in stimulated saliva is rejected.

## 4. Materials and Methods

### 4.1. Study Participants

In this study, we enrolled patients with ill-fitting dentures who were diagnosed during their first visit at Kanagawa Dental University Hospital as requiring new dentures for occlusion treatment. Participants excluded from this study include the following criteria: participants with dementia, psychiatric disorders that may interfere with communication, moderate or severe periodontal disease, organic diseases related to salivary glands, and autoimmune diseases as typified by Sjogren’s syndrome were excluded. Participants with symptoms ofdry mouth, which makes saliva collection difficult, were excluded because at least 1 mL must be collected of saliva samples used in the analysis. Participants who smoked and previously smoked were excluded to rule out the effects of smoking on salivary glands and salivary secretion. Participants with a systemic medical history and taking regular medication were: hypertension: 4 subjects; diabetes: 1 subject; dyslipidemia: 1 subject; osteoporosis: 1 subject; no systemic medical history: 14 subjects. Participants had no change in their medications during this study period, nor any sudden changes in their medical conditions. Prosthodontic treatment was restricted to selected prosthodontic dentists with more than 5 years of experience. This study was limited to participants without dentures and without occlusal support by natural teeth to identify changes in salivary metabolites due to denture treatment. Eichner’s classification of missing dentition was used to record oral status. The classifications were as follows: C1; remaining teeth in the upper and lower jaws but no interocclusal relationship, C2; edentulous in a single jaw, and C3; edentulous in both the upper and lower jaws. The sample size was calculated using G*Power 3.1.9.3 with a statistical power of 0.8 and an effect size of 0.5. As a result, 21 participants (8 men and 13 women; age, 78.2 ± 8.4 years) were recruited (Table 1). None of the participants discontinued treatment during the study period. This study was approved by the Ethics Committee of Kanagawa Dental University (approval no. 891, issued on 1 June 2016), and informed consent was obtained from all participants.

### 4.2. Measurement Items

All examinations were performed twice, once at the initial visit before denture treatment and twice after denture treatment was completed. After the new denture was fitted and adjusted by the prosthodontist, the patient’s discomfort is confirmed to have disappeared. Furthermore, the denture was diagnosed as complete when the prosthodontist diagnosed that the denture was stable approximately three months later.

#### 4.2.1. Salivary Flow Rates

Saliva (first resting saliva, then stimulated saliva) was collected in the morning (9:00–11:00) on either of the first two days of the week. Provision was made to minimize the effects of diurnal and inter-day variability on salivary flow and salivary metabolites [38]. Swallowing was prohibited during the sampling procedure. The saliva sampling conditions were unified in terms of sampling method, sampling location, and not eating or drinking anything other than water within 2 h before saliva sampling. All specimens were collected and measured by the same researcher. Resting saliva was obtained by collecting saliva using the spitting method. Saliva was collected in a Falcon^®^ tube (Corning Inc., Tokyo, Japan) placed on the lower lips. The total amount of resting saliva sampled over 15 min [19] was measured; the reference normal value was ≤1.5 mL/15 min [39]. Similarly, stimulated saliva was collected while chewing gum (FREE ZONE, Lotte, Tokyo, Japan). The total amount of stimulated saliva sampled over 10 min [19] was measured; the reference normal value was ≤10 mL/10 min [30]. All collected saliva samples were stored frozen at −20 °C.

#### 4.2.2. Masticatory Ability Test

To confirm the effect of the denture treatment, mastication efficiency was assessed using a masticatory ability test device (Glucosensor GS-II; GC, Tokyo, Japan) to objectively evaluate the treatment-induced changes. Glucose-containing gums were chewed for 20 s on the main chewing side. Then, the mouth was washed with 10 mL of water that was expectorated into a container through a filter mesh. The filtrate dropped onto a sensor chip of the device, and the glucose concentration of the filtrate was recorded in mg/dL. The reference value for the measured value was ≥100 mg/dL [40].

### 4.3. Metabolome Analysis

Frozen saliva was thawed, passed through a 5 kDa cutoff filter (Nihon Pall, Ltd., Tokyo, Japan), centrifuged for at least 2.5 h at 9100× *g* and 40 °C, and filtered to remove high-molecular-weight compounds. Next, 5 μL of Milli-Q water (Millipore Corporation, Bedford, MA, USA) containing methionine sulfone, 2-[N-morpholino]-ethanesulfonic acid, D-camphor-10-sulfonic acid, 3-amino pyrrolidine, and trimesate, each at 2 mmol/L, was added to 45 μL of the filtrate, followed by mixing. The processed samples were analyzed using CE-TOFMS in positive and negative modes. MasterHands (version 2.18.1.2, Keio University, Yamagata, Japan) was used to analyze these CE-TOFMS data. A noise filtering sub was used for noise filtering, baseline subtraction, peak integration of each sliced electropherogram, and accurate *m/z* estimation of the mass spectrum. Estimation of mass spectrum *m/z*, generation of the peak matrix by aligning multiple datasets, and identification of each peak by *m/z* matching were performed. Each peak was identified by matching the *m/z* value and corrected migration time to the corresponding entry in the standard library. The metabolite concentrations were calculated based on the ratio of the peak area to the area of the internal standard in the samples and standard compound mixtures.

### 4.4. Data Analysis

Continuous data are presented as mean ± standard deviation. Changes in secretion volumes of resting and stimulated saliva, as well as changes in masticatory efficiency before and after treatment, were statistically analyzed using Wilcoxon’s matched-pairs signed-rank test. The significance level was set to 0.05. All statistical data analyses were performed using XLStat (version 2014.1.09, XLStat, Paris, France), R software (version 2.14.0, R Foundation for Statistical Computing, Vienna, Austria, https://www.r-project.org/; accessed on 1 August 2022), and GraphPad Prism (version 5.0.2 GraphPad Software Inc., San Diego, CA, USA).

For data visualization, a heatmap, principal component analysis (PCA), and volcano plot were created using MetaboAnalyst (ver. 5.0; https://www.metaboanalyst.ca/; accessed on 1 August 2022).

## 5. Conclusions

In this study, we investigated the effects of denture treatment on salivary metabolites. Four salivary metabolites (N1-acetyl spermidine, betaine, malate, and 2-hydroxy-4-methyl pentanoate) had significantly decreased concentrations in both resting and stimulated saliva after denture treatment. These study findings suggest that the improvement in masticatory function by dentures not only affects the amount of secreted saliva but also the composition of salivary metabolites.

## Figures and Tables

**Figure 1 ijms-24-13959-f001:**
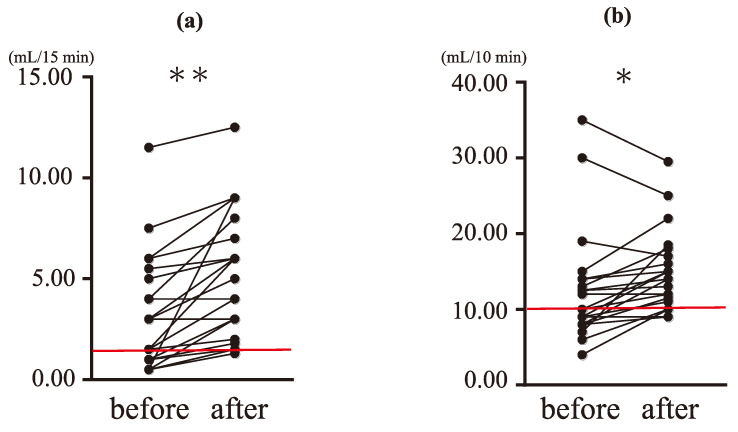
Changes in resting and stimulated salivary secretion before and after treatment. (**a**) Resting salivary volume (mL/15 min). (**b**) Stimulated salivary volume (mL/10 min). The reference normal values were indicated by red lines. The reference values are ≤1.5 mL/15 min (resting saliva test) and ≤10 mL/10 min (stimulated saliva test). Many participants whose values were below the reference value before the treatment showed an improvement. Stimulated salivary volume decreased in some participants in the range above the reference value after treatment. Wilcoxon’s matched-pairs signed-rank test (* *p* < 0.05, ** *p* < 0.0001).

**Figure 2 ijms-24-13959-f002:**
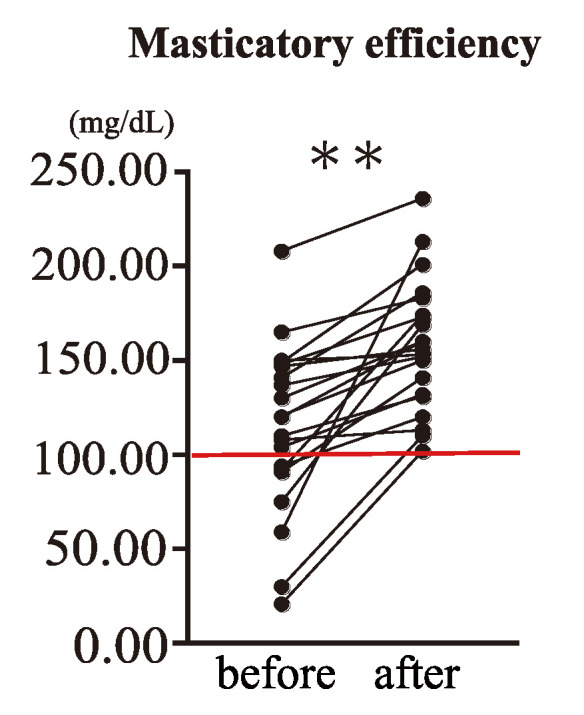
Masticatory efficiency before and after denture treatment. The reference value for the measured value was 100 mg/dL or higher, indicated by the red line. All participants showed an increase in masticatory efficiency after the treatment, equal to or exceeding the reference values. Wilcoxon’s matched-pairs signed-rank test (** *p* < 0.0001).

**Figure 3 ijms-24-13959-f003:**
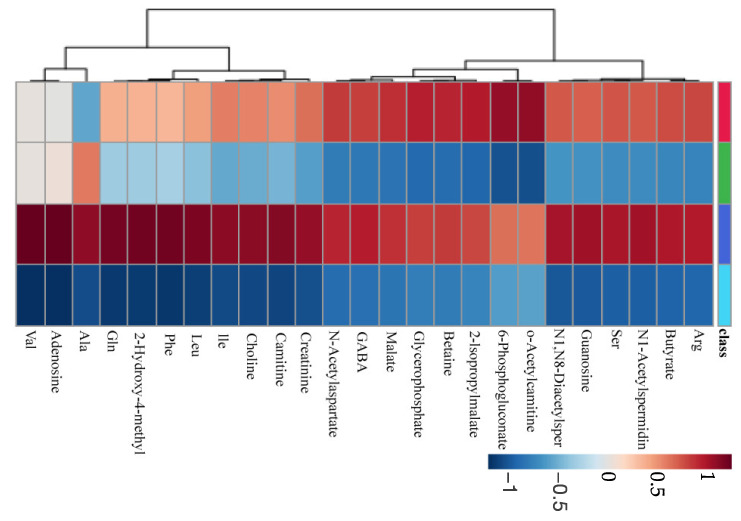
Heatmap of metabolite concentrations in resting and stimulated saliva before and after treatment. Metabolites detected in more than 50% of all samples are visualized, and 25 metabolites with large concentration changes are shown. The average value of each metabolite was calculated separately for resting and stimulated saliva, and the ratio was calculated by dividing each by the average value. Low- and high-concentration metabolites are indicated in blue and red, respectively. Before and after treatment classified for the saliva of each is indicated. Class categories: red (resting saliva before-treatment), green (resting saliva after-treatment), blue (stimulated saliva before-treatment), and light blue (stimulated saliva after-treatment). Clustering by Pearson’s correlation coefficient.

**Figure 4 ijms-24-13959-f004:**
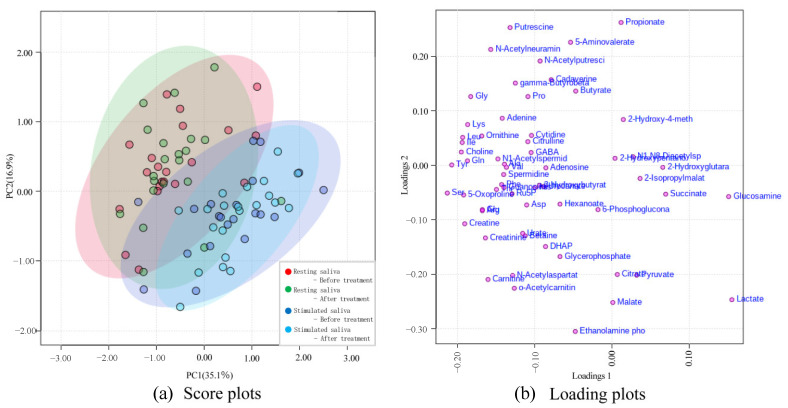
Principal component analysis (PCA) of salivary metabolites. (**a**) Score plots. The contributions to the principal components (PCs) are 35.1% on the horizontal axis (PC1, reflecting the overall concentration in saliva samples) and 16.9% on the vertical axis (PC2, the second-highest PC). Red: resting saliva before treatment, green: resting saliva after treatment, blue: stimulated saliva before treatment, light blue: stimulated saliva after treatment. (**b**) Loading plots. Many metabolites can be observed in regions with PC1 < 0. The contribution rate of PC1 (35.1%) is higher than that of PC2 (16.9%).

**Figure 5 ijms-24-13959-f005:**
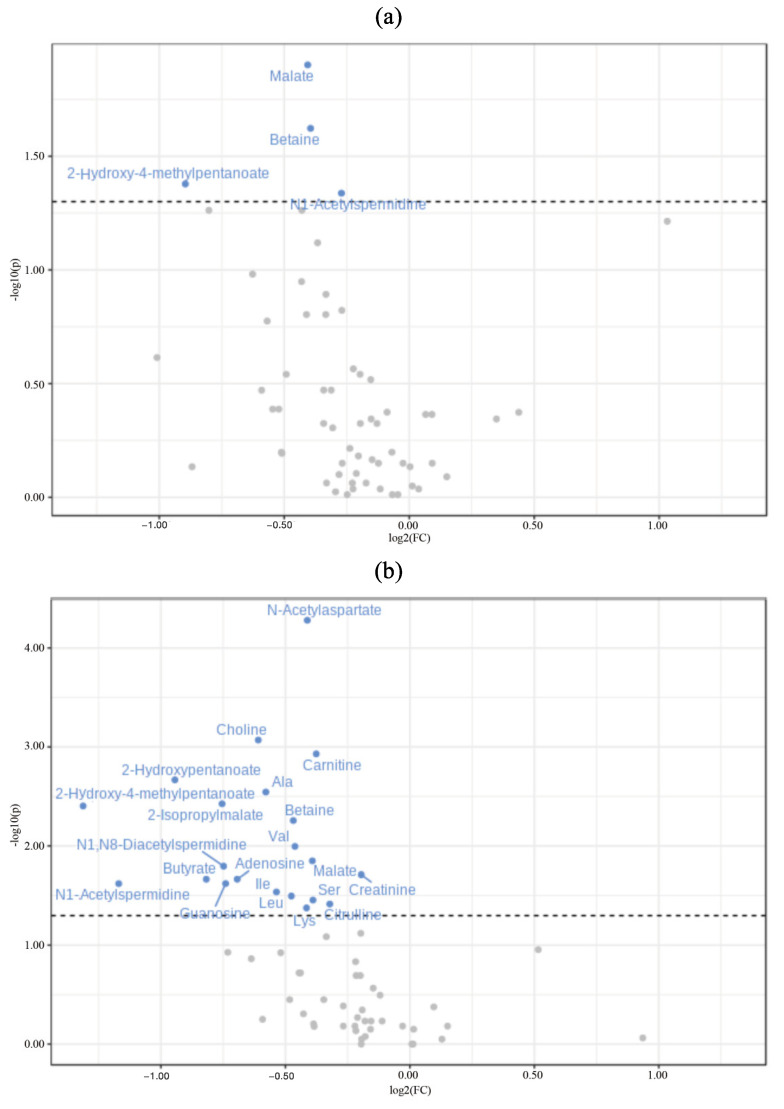
Volcano plot to show changes in metabolite concentrations before and after treatment. (**a**) Resting saliva. (**b**) Stimulated saliva. The ratio of salivary metabolite concentrations before and after treatment is shown. The horizontal axis indicates the rate of change. Metabolites shown to the left of 0 indicate substances with decreased concentrations after treatment, and metabolites shown to the right indicate those with increased concentrations. The vertical axis represents the *p*-value. Metabolites plotted above 1.3 are those with significant differences. As most of the plots are to the left of 0, most metabolite concentrations tended to decrease after treatment.

**Table 1 ijms-24-13959-t001:** Subject profile.

Age		N	Ave	SD (Standard Deviation)
Total		21	78.2	8.4
Eichner Index	C1	3		
	C2	9		
	C3	9		
Smoking	Yes	0		
	No	21		
Medical disease	Hypertension	4		
	Diabetes mellitus	1		
	Hyperlipidemia	1		
	Osteoporosis	1		
	Other (No medicine)	1		

## Data Availability

All the raw data elaborated in this study are provided in Appendix A.

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
