# Peer review of "Effects of Denture Treatment on Salivary Metabolites: A Pilot Study"

_ijms, 2023, doi:10.3390/ijms241813959_

Round 1
Reviewer 1 Report
The manuscript entitled ‘Effects of Denture Treatment on Salivary Metabolites: A Pilot Study’ investigated the effects of denture treatments on salivary secretion volume and salivary metabolite composition.
The findings presented in the manuscript possess a predominantly descriptive nature, lacking a more profound analysis regarding the mechanisms underlying how and why denture treatments could induce alterations in salivary metabolite composition.
Furthermore, the salivary metabolite composition and secretion might be significantly influenced by the systemic diseases that the participants may have. Therefore, establishing a definite connection between denture treatment and alterations in salivary secretion volume and metabolite composition was challenging. The authors should have rigorous criteria for participant selection.
Additionally, there are a couple of supplementary points to address:
In Figure 3, there is a duplication of the label 'Stimulate saliva-before treatment' in the figure legend.
The quality of resolution in Figure 4 and Figure 5 is notably insufficient, leading to difficulties in reading and interpreting the content.
The English language is fine.
Author Response
Dear reviewer 1
We wish to express our strong appreciation to the Reviewer for insightful comments on our paper. We feel the comments have helped us significantly improve the paper. We have made changes to the points you pointed out in red.Thank you very much for your cooperation.
The findings presented in the manuscript possess a predominantly descriptive nature, lacking a more profound analysis regarding the mechanisms underlying how and why denture treatments could induce alterations in salivary metabolite composition.
→We wish to thank the Reviewer for this comment. We have revised Discussion . (lines 191-263)
Furthermore, the salivary metabolite composition and secretion might be significantly influenced by the systemic diseases that the participants may have. Therefore, establishing a definite connection between denture treatment and alterations in salivary secretion volume and metabolite composition was challenging. The authors should have rigorous criteria for participant selection.
→We wish to thank the Reviewer for this comment. We have added Materials and Methods to the Study Participants. (lines266-290)
In Figure 3, there is a duplication of the label 'Stimulate saliva-before treatment' in the figure legend.
→We appreciate the Reviewer's comment on this point. We have Changed the legend in fig3.
The quality of resolution in Figure 4 and Figure 5 is notably insufficient, leading to difficulties in reading and interpreting the content.
→We appreciate the Reviewer's comment on this point. We have changed fig4 and fig5 to higher resolutions.
Thank you again for your comments on our paper. We trust that the revised manuscript is suitable for publication.
Reviewer 2 Report
Dear authors,
Thank you for submitting your valuable work to the journal. The loss of teeth triggers a significant decrease of the patients' life quality. Hence, the use of dentures to restore the dental arches is still recommended in patient who cannot afford or have contraindications for implant therapy. Thus, the topic of your paper is interesting and of significant clinical value.
However I would suggest some changes to the manuscript in order to improve its scientific accuracy.
- Please provide a Null Hypothesis along the Objectives of your study
- Please expand on the inclusion/exclusion criteria of the selected patients
- Please provide higher resolution images for Figures 4 and 5
- Please include more relevant references in the Discussions
- Please rephrase Conclusions, for increased comprehensibility
We look forward to receiving the revised version of your manuscript
Kind regards
Minor check-up
Author Response
Dear reviewer 2
We wish to express our strong appreciation to the Reviewer for insightful comments on our paper. We feel the comments have helped us significantly improve the paper. We have made changes to the points you pointed out in red. Thank you very much for your cooperation.
Please provide a Null Hypothesis along the Objectives of your study
→We appreciate the Reviewer's comment on this point. We have revised Introduction. (lines 98-100)
Please expand on the inclusion/exclusion criteria of the selected patients.
→We wish to thank the Reviewer for this comment. We have added Materials and Methods to the Study Participants. (lines269-293)
Please provide higher resolution images for Figures 4 and 5.
→We appreciate the Reviewer's comment on this point. We have changed fig4 and fig5 to higher resolutions.
Please include more relevant references in the Discussions.
→We wish to thank the Reviewer for this comment. We have added references to the following parts : lines;192-265.
Please rephrase Conclusions, for increased comprehensibility
→We wish to thank the Reviewer for this comment. We have changed conclusion.
Thank you again for your comments on our paper. We trust that the revised manuscript is suitable for publication.
Round 2
Reviewer 1 Report
The authors have addressed my concerns in my previous report. I have no further questions.
Minor editing of the English language would be beneficial.